# The Safety INdEx of Prehospital On Scene Triage (SINEPOST) study: The development and validation of a risk prediction model to support ambulance clinical transport decisions on-scene

**Jamie Miles**[1]*, **Richard Jacques**[2], **Richard Campbell**[1], **Janette Turner**[1], **Suzanne Mason**[1]

**1** Centre for Urgent and Emergency Care, School of Health and Related Research, The University of Sheffield, Sheffield, United Kingdom, **2** Design, Trials and Statistics, School of Health and Related Research, The University of Sheffield, Sheffield, United Kingdom

* j.miles@sheffield.ac.uk

**Data Availability Statement:** This study used patient sensitive linked data from two sources. Yorkshire Ambulance Service NHS Trust and NHS

## Abstract

One of the main problems currently facing the delivery of safe and effective emergency care is excess demand, which causes congestion at different time points in a patient's journey. The modern case-mix of prehospital patients is broad and complex, diverging from the traditional 'time critical accident and emergency' patients. It now includes many low-acuity patients and those with social care and mental health needs. In the ambulance service, transport decisions are the hardest to make and paramedics decide to take more patients to the ED than would have a clinical benefit. As such, this study asked the following research questions: In adult patients attending the ED by ambulance, can prehospital information predict an avoidable attendance? What is the simulated transportability of the model derived from the primary outcome? A linked dataset of 101,522 ambulance service and ED ambulance incidents linked to their respective ED care record from the whole of Yorkshire between 1st July 2019 and 29th February 2020 was used as the sample for this study. A machine learning method known as XGBoost was applied to the data in a novel way called Internal-External Cross Validation (IECV) to build the model. The results showed great discrimination with a C-statistic of 0.81 (95%CI 0.79–0.83) and excellent calibration with an O: E ratio was 0.995 (95% CI 0.97–1.03), with the most important variables being a patient's mobility, their physiological observations and clinical impression with psychiatric problems, allergic reactions, cardiac chest pain, head injury, non-traumatic back pain, and minor cuts and bruising being the most important. This study has successfully developed a decision-support model that can be transformed into a tool that could help paramedics make better transport decisions on scene, known as the SINEPOST model. It is accurate, and spatially validated across multiple geographies including rural, urban, and coastal. It is a fair algorithm that does not discriminate new patients based on their age, gender, ethnicity, or decile of deprivation. It can be embedded into an electronic Patient Care Record system and automatically calculate the probability that a patient will have an avoidable attendance at the ED,

Digital. The latter source collects routine healthcare data from all hospital trusts in England and compiles them into data products. The data products used in this study were the Emergency Care Data Set (ECDS) and the Hospital Episode Statistics Accident and Emergency (HES AE). More information on these data products can be found here: https://digital.nhs.uk/services/data-access-request-service-dars/dars-products-and-services. The minimum data set for this study is unavailable because access to both sources of data was through two data sharing agreements to process identifiable information, link the records and to analyse the data in a specific way. The data sharing agreements extended to state that the data must be destroyed once all analysis had been completed on the data. For reproducing the research, the parameters of records for Yorkshire Ambulance Service NHS Trust and NHS Digital have been outlined in the above sections. It is possible for future researchers to request the same datasets from these providers, and the researchers in this study had no special access privileges to the data. Please contact yas.research@nhs.net or enquiries@nhsdigital.nhs.uk for more information. S1 Appendix in the supporting information shows a detailed flow diagram of the data processing and linking.

**Funding:** JM ICA-CDRF-2018-04-ST2-044 National Institute for Health and Care Research (NIHR) https://www.nihr.ac.uk/explore-nihr/academy-programmes/hee-nihr-integrated-clinical-and-practitioner-academic-programme.htm he funders had no role in study design, data collection and analysis, decision to publish, or preparation of the manuscript.

**Competing interests:** The authors have declared that no competing interests exist.

if they were transported. This manuscript complies with the Transparent Reporting of a multivariable prediction model for Individual Prognosis Or Diagnosis (TRIPOD) statement (Moons KGM, 2015).

# Background

In the emergency care system, pressure is rising amidst the growing quantity of patients accessing front door services such as the ambulance service, Emergency Department (ED) and General Practice (GP). This demand is rising at around 5% per annum [2, 3]. For the ambulance service, this means that patients who are transported to hospital may be held in a queue of other ambulances waiting to hand their patients over. In 2019/2020 in England alone there were 137,009 delays in ambulance handover of between 30 and 60 minutes [4]. When these delays occur and ambulances are queueing, it has the potential to cause harm to those in the queue. A recent report from the Association of Ambulance Chief Executives (AACE) found that 80% of ambulance patients that queued for more than an hour experienced some level of harm [5]. Studies have been more specific in identifying harm that has occurred with certain diseases. It has been shown that delayed handover in patients with non-traumatic chest pain is associated with a greater risk of 30-day mortality [6]. There are also potential consequences for prehospital patients still waiting to be assessed in the community.

The case mix of these patients is not always life-threatening emergencies. Previous reports have demonstrated that the majority of prehospital patients have no immediate life-threatening care need and their actual need could be managed in the community [7, 8]. However, some of these patients are still transported to the ED and this can lead to an avoidable ED attendance.

When paramedics make decisions on-scene to transport a patient to hospital, it is often the most complex decision they make [9]. As such, the decision is not always accurate. Studies have found that there are between 9 and 32% of ambulance transports to ED that could have been avoided [7, 10–12]. It is recognised that in some systems transport decisions are not clinician-made and patient-centred, but financially driven through payment policies [13, 14]. However, these policies are beginning to adapt to the modern case-mix of patients and as such, the adoption of a transport decision support tool would be of high benefit and importance.

Existing transport decision support tools that are in practice have all been designed not to miss a higher acuity patient, which has led to significant over-triage of patient acuity. They have also failed to demonstrate significant benefit over clinician decision making. A vignette-based survey by Miles et al. found that conveyance decisions had a sensitivity of 0.89 (95% CI 0.86–0.92) and a specificity of 0.51 (95% CI 0.46–0.56) [15]. This is comparable to existing decision support tools such as the paramedic pathfinder [16, 17]. A systematic review into whether machine learning computerised decision support could offer an improvement on triage found that certain methods such as decision trees, neural networks and logistic regression all were able to provide accurate discrimination between different acuity levels. A limitation of the included studies was that they were often predicting high acuity [18].

If current clinical judgement is already sensitive to identifying high-acuity patients, the benefit of a decision support tool is on triaging the mid- and low-acuity. If accuracy is improved at this level of triage, the benefit would be a reduction in the avoidable transportation of patients to an ED.

## Objectives

**Primary research question.** In adult patients attending the ED by ambulance, can prehospital information predict an avoidable attendance?

**Primary objectives.**

1) Extract prehospital variables from ambulance service electronic patient care records

2) Link the data with ED electronic patient care records

3) Identify low acuity patients in the dataset using the ED information

4) Build a predictive model using prehospital variables

5) Measure the success of the model in predicting an avoidable attendance using prehospital variables.

**Secondary research questions.** What is the simulated transportability of the model derived from the primary outcome?

**Secondary objectives.**

6) Test spatial validation

7) Test model discrimination of protected characteristics

## Methods

### Source of data

This retrospective cohort study analysed a sample of ambulance service patients transported to the ED between the 1st July 2019 and the 29th February 2020. Each episode had an ambulance electronic Patient Care Record (ePCR) created which contained all demographic and clinical information. The Yorkshire Ambulance Service (YAS) provided this data. The outcome was generated using two ED-based data products from NHS Digital, which were then subsequently linked to the ambulance data. The two products were the Hospital Episode Statistics Accident and Emergency (HES A&E) and the Emergency Care Data Set (ECDS).

### Participants

In this study, all patients who were over the age of 18 that had a face-to-face paramedic contact from Yorkshire Ambulance Service (YAS) with a completed ePCR were eligible for inclusion. For the development of the prediction model, each transported instance was linked to its respective ED record. A total of 17 EDs were included in the study and a full list of these can be found in S2 Appendix. The patients were not selected by any specific demographic or disease. This was to ensure the model could be applied to all patients. Children were excluded from the model as they are a cohort who are confounded by ambulance service policy.

### Outcome

The outcome is an avoidable conveyance attendance at the ED, which is an experienced based definition initially described by O'Keeffe et al. as "first attendance with some recorded treatments or investigations all of which may have reasonably been provided in a non-emergency care setting, followed by discharge home or to GP care" [12]. This was operationalised into a data-driven definition and can be found in the protocol publication [18].

## Predictors

All candidate variables were measured whilst the ambulance crew was with the patient pro-spectively. Data was retrieved after the data collection period, and no ambulance crew was aware of the study during data collection. Variables can be broadly categorised into demo-graphic, clinical, social, and interventional. S7 Appendix displays all candidate variables, exam-ple values, justification for inclusion and assigned parameters within each variable. The only demographic variable included was incident location as a categorical variable. Age was initially included however, after initial model building it was found to introduce a bias and was removed. Incident location is user inputted by the ambulance crew depending on whether the patient is at a domestic address, public place, care home, work or other. Clinical variables formed most of the candidate variables. When a paramedic arrives on scene, they will first undertake a primary survey. This records whether the patient has a catastrophic haemorrhage, if their airway is clear, if they are breathing normally, or if there are any obvious circulation issues. These are all recorded as categorical variables. The patient will then have physiological variables recorded to assess how serious their medical complaint may be. Pulse rate is mea-sured in beats per minute (bpm) and is the frequency at which the heart beats in a minute. Tra-ditionally this is measured by palpation of the pulse, however technology allows this to be measured using medical equipment. Respiratory rate is measured as respirations per minute (rpm) and is a manual count of the number of breaths the patient takes in one minute. Tem-perature is a continuous variable measured in ˚C using a tympanic thermometer. The periph-eral capillary oxygen saturation in the blood (SpO2) is measured using medical equipment as a percentage. Blood sugar levels are also recorded using a machine that takes a small blood sam-ple. The results are recorded as mmol per litre. Blood pressure is recorded using millimetres of mercury (mmHg). Two measurements are recorded, the systolic blood pressure and the dia-stolic blood pressure. The level of consciousness is calculated using a four-scale system (AVPU) in the primary survey and the Glasgow Coma Scale (GCS) in the physiological obser-vations. GCS is a composite score of labelled scales. The minimum score is three and maxi-mum fifteen [19]. Baseline oxygen demands, and current oxygen demands are recorded as binary variables. All the physiological variables are combined to calculate a National Early Warning 2 score (NEWS2) [20, 21]. The NEWS2 score has been included as a candidate pre-dictor and treated as categorical. The NEWS2 assigns points between 0 and 3 to physiological variables depending on how deranged the values are. The minimum NEWS2 score is 0, and the maximum is 20 [21]. Clinical variables include pain scores out of ten, subsequent measure-ments of observations and feature engineered intervals between primary measurements and subsequent ones. All sixteen clinical interventions (e.g., cannulation, intubation etc.) were included as binary variables. The patient's mobility was recorded depending on what resource they required, i.e., self-mobile, stretcher needed, carry chair needed etc. This variable was how the patient was able to move to and from the ambulance and was a categorical variable. Clinical impression was also included as a categorical variable with 99 different values to possibly select. Examples include 'head injury', 'shortness of breath', and 'abdominal pain'. Social vari-ables were included as binary variables. These were included as surrogates to determine the level of external support the patient has. These include variables such as GP details recorded, social worker recorded etc. It also included referral variables if the patient was referred to a ser-vice such as falls, safeguarding or diabetes clinic etc.

## Sample size

The sample size was calculated using the 'pmsampsize v1.1.0' for R v3.6.1 for windows [22]. Two studies by Riley et al. also informed the sample size calculation [23, 24]. Previous studies

have found a conservative estimate of the outcome prevalence to be 0.085 [12]. A meta-analysis found that the average C-statistic was 0.8 [25]. A preliminary analysis of a separate dataset found that there was potentially 637 parameters in the ambulance service dataset. This gave an estimated sample size of 55,676 with an anticipated 4733 event and an events per parameter (EPP) of 7.43.

## Missing data

The strategy for handling missing data was to first elicit if missing values in each variable were the negative class. For example, the clinical procedure of intravenous cannulation is only recorded in the ePCR if the patient was cannulated. Therefore, it is logical, in the absence of a positive recording to assume the patient was not cannulated and the missing data can be transformed into the negative class. Once this has been completed, any variable with more than 30% missing data was excluded from the analysis. The rationale for this is that it may not be routinely, or accurately completed in the ePCR and to include them could lead to model failure in practice.

## Statistical analysis methods

The full statistical analysis plan has been published in the study protocol [18]. In this study, an XGBoost algorithm was used for model development. Recursive feature elimination was used to subset the candidate variables into only the most important that provided the most accurate prediction model. Then the algorithms hyperparameters were tuned to prevent model overfitting. The model was first evaluated for its calibration using Spiegelhalter's Z-test. Then, model discrimination was assessed using the C-statistic (area under the ROC curve). The optimal threshold was identified by finding the closest top left point of the ROC curve. This was then used to assess accuracy statistics. Once the full model was completely developed, symmetrical procedures were undertaken using different Emergency Departments as held-out test sets with all remaining data as the training data. This in effect created a full model and seventeen other models which could then be meta-analysed. The summary statistics generated in a random effects meta-analysis were then used to update the final model for its performance. In the protocol paper, the full procedures are outlined in detail [18]. This study is a development study with internal-external validation using a meta-analysis of ED clusters. There is no external validation.

**Data linkage and dataset creation.** YAS identified and extracted all eligible ePCR records from its information system between 1st July 2019 to 29th February 2020. These dates were bound by two time points. The 1st of July 2019 was when YAS launched the regional role out of the ePCR. The 29th of February 2020 was the last date possible, before the COVID-19 pandemic would confound the sample. This extract was partitioned into two datasets: one that included identifiable fields but no clinical fields, which was transferred to NHS Digital; and a second composed of the same records with clinical data (directly identifiable fields removed), that was transferred to the University of Sheffield project team. Both datasets contained a common identifier field to enable linkage. NHS Digital attempted to trace patients' identities based on the combinations of identifiers they received from YAS. Records for the cohort successfully traced by NHS Digital were extracted from the requested datasets (HES A&E and ECDS) and sent to the project team. Previous data linkage methodology with NHS Digital used an eight stage hierarchical probabilistic matching algorithm [26]. However, the ECDS data product could only be linked using the unique identifier of NHS number, which renders the linkage process to be largely deterministic. As a result, all patient records sent to NHS Digital with an NHS number were successfully linked, whereas those without an NHS number were not. This

resulted in 195078 (66%) of the total cohort excluded from the analysis. A comparison of the successfully linked cohort and the unlinked cohort revealed no fundamental differences.

YAS and NHS Digital both removed records that belonged to patients who had registered an NHS national data opt-out. Duplicate records were also removed from both datasets to ensure that a single person's records did not appear more than once.

All three received datasets (YAS ePCR, HES A&E, and ECDS) were linked using a consistent patient-level identifier. For ambulance incidents linked to HES A&E attendances only the earliest A&E attendance record with a datetime after the latest (by datetime) ambulance incident datetime and no more than 6 hours later were retained. This ensured that the link between an ambulance incident and HES A&E record remained one-to-one.

To link HES A&E data to ECDS data (and therefore ECDS to ambulance incidents), it was chosen to link via the common identifier—arrival time pairs. If there were multiple records from this linkage, the "most complete" record was chosen. The most complete was determined by the presence of fields that are used to calculate if an attendance is of low acuity. A graphical representation of data flow and linkage can be found in S1 Appendix.

**Ethics statement.**   This study underwent extensive ethical review. It was first reviewed and approved by the South Yorkshire NHS Research Ethics Committee (REC) on the 20[th] December 2019. It was also reviewed and approved by the NHS Confidentiality Advisory Group (CAG) on the 14[th] July 2020. During the data sharing agreement stage, it was further reviewed and approved by the NHS Digital Independent Group Advising on the Release of Data (IGARD) team on the 15[th] Feb 2021. This study used patient data without written or verbal patient consent as it was not feasible to achieve this with the large volume of retrospective data. To mitigate this, the patient identifiers were first screened against the NHS National data opt out. This removed all patient episodes where the patient had previously stated they did not want their data used for the purposes of research. To further mitigate this, privacy notices were shared on both the Yorkshire Ambulance Service NHS Trust and the University of Sheffield websites. These contained contact details to remove participants from the study, prior to pseudonymisation.

## Results

### Participants

There were 101,522 individual patient episodes included in the analysis. Of these, 7228 (7.12%) were defined as having an avoidable ambulance conveyance to the ED. Table 1 provides key demographic information between those with, and without the outcome. It also shows physiological observations as a surrogate for comparative patient acuity. In the supplementary material, the table is extended to show the clinical impression fields.

### Model development

**Dataset preparation.**   During the preparation of the dataset there were 215 possible candidate variables for inclusion which comprised of 190 categorical variables (including 169 binary variables), and 25 continuous variables. After one hot encoding there were 452 candidate predictors in the final dataset. During recursive feature elimination, the ideal set of variables was found to be only 90 of the total candidate variables. These condensed down into 19 variables, comprising of 14 clinical variables, 3 interventional and 2 demographics. A full list of included candidate variables can be found in S3 Appendix

**Model performance.**   In an XGBoost algorithm, the hyperparameters that control how the model is built prevents the model overfitting the training data. Therefore the apparent validity

**Table 1. Characteristics of participants.**

| | Unavoidable | Avoidable | Overall |
|---|---|---|---|
| | (N = 94294) | (N = 7228) | (N = 101522) |
| **Gender** | | | |
| Female | 52620 (93%) | 4120 (7%) | 56740 |
| Male | 41572 (93%) | 3100 (7%) | 44672.00 |
| Transgender | 7 (88%) | 1 (13%) | 8 |
| Unknown | 95 (93%) | 7 (7%) | 102 |
| **Age** | | | |
| Mean (SD) | 66.8 (20.3) | 50.9 (22.6) | 65.7 (20.9) |
| Median [Min, Max] | 72.0 [18.0, 107] | 48.0 [18.0, 107] | 71.0 [18.0, 107] |
| **Ethnicity** | | | |
| African (Black or Black British) | 269 (89%) | 33 (11%) | 302 |
| Caribbean (Black or Black British) | 380 (91%) | 36 (9%) | 416 |
| Any other Black background | 164 (86%) | 26 (14%) | 190 |
| Bangladeshi (Asian or Asian British) | 124 (84%) | 23 (16%) | 147 |
| Chinese (Asian or Asian British) | 59 (86%) | 10 (14%) | 69 |
| Indian (Asian or Asian British) | 521 (90%) | 55 (10%) | 576 |
| Pakistani (Asian or Asian British) | 2894 (87%) | 415 (13%) | 3309 |
| Any other Asian background | 382 (85%) | 67 (15%) | 449 |
| British (White) | 78401 (94%) | 5420 (6%) | 83821 |
| Irish (White) | 361 (93%) | 29 (7%) | 390 |
| Any other White | 2464 (90%) | 263 (10%) | 2727 |
| White and Asian (Mixed) | 76 (85%) | 13 (15%) | 89 |
| White and Black African (Mixed) | 35 (81%) | 8 (19%) | 43 |
| White and Black Caribbean (Mixed) | 113 (92%) | 10 (8%) | 123 |
| Any other Mixed background | 150 (90%) | 17 (10%) | 167 |
| Any other ethnic group | 761 (84%) | 142 (16%) | 903 |
| Unknown | 2554 (90%) | 281 (10%) | 2835 |
| Not stated | 4586 (92%) | 380 (8%) | 4966 |
| **Incident location** | | | |
| Care Home | 7614 (95%) | 372 (5%) | 7986 |
| Domestic Address | 68004 (93%) | 5281 (7%) | 73285 |
| Not Selected | 27 (96%) | 1 (4%) | 28 |
| Other | 4449 (93%) | 320 (7%) | 4769 |
| Public Place | 2710 (89%) | 335 (11%) | 3045 |
| School | 30 (70%) | 13 (30%) | 43 |
| Work | 473 (88%) | 65 (12%) | 538 |
| Missing | 10987 (93%) | 841 (7%) | 11828 |
| **Transported ED** | | | |
| Airedale General Hospital | 3058 (93%) | 240 (7%) | 3298 |
| Barnsley District General | 5810 (95%) | 323 (5%) | 6133 |
| Bradford Royal Infirmary | 6705 (87%) | 1004 (13%) | 7709 |
| Calderdale Royal Hospital | 3865 (94%) | 242 (6%) | 4107 |
| Dewsbury District Hospital | 827 (86%) | 137 (14%) | 964 |
| Doncaster Royal Infirmary | 6258 (94%) | 420 (6%) | 6678 |
| Harrogate District Hospital | 2598 (94%) | 163 (6%) | 2761 |
| Huddersfield Royal Infirmary | 4392 (94%) | 283 (6%) | 4675 |
| Hull Royal Infirmary | 10099 (94%) | 612 (6%) | 10711 |

(*Continued*)

**Table 1.** (*Continued*)

| | Unavoidable | Avoidable | Overall |
|---|---|---|---|
| | (N = 94294) | (N = 7228) | (N = 101522) |
| James Cook University Hospital | 749 (93%) | 55 (7%) | 804 |
| Leeds General Infirmary | 4839 (95%) | 263 (5%) | 5102 |
| Northern General Hospital | 9793 (91%) | 929 (9%) | 10722 |
| Pinderfields General Hospital | 9481 (93%) | 764 (7%) | 10245 |
| Rotherham District General Hospital | 5618 (94%) | 352 (6%) | 5970 |
| Scarborough District General Hospital | 4374 (97%) | 120 (3%) | 4494 |
| St James University Hospital | 8078 (91%) | 824 (9%) | 8902 |
| York District Hospital | 5719 (94%) | 382 (6%) | 6101 |
| Missing | 2031 (95%) | 115 (5%) | 2146 |
| **Indices of Deprivation** | | | |
| 1 | 22882 (91%) | 2331 (9%) | 25213 |
| 2 | 12177 (92%) | 1054 (8%) | 13231 |
| 3 | 9934 (92%) | 817 (8%) | 10751 |
| 4 | 7439 (93%) | 518 (7%) | 7957 |
| 5 | 7560 (94%) | 484 (6%) | 8044 |
| 6 | 8025 (94%) | 504 (6%) | 8529 |
| 7 | 7801 (94%) | 459 (6%) | 8260 |
| 8 | 7199 (94%) | 432 (6%) | 7631 |
| 9 | 5959 (95%) | 346 (5%) | 6305 |
| 10 | 5199 (95%) | 272 (5%) | 5471 |
| Missing | 119 (92%) | 11 (8%) | 130 |
| **Initial Pulse rate (bpm)** | | | |
| Mean (SD) | 89.2 (22.3) | 88.1 (18.2) | 89.1 (22.0) |
| Median [Min, Max] | 86.0 [5.00, 220] | 87.0 [6.00, 220] | 86.0 [5.00, 220] |
| Missing | 2186 (2.3%) | 309 (4.3%) | 2495 (2.5%) |
| **Initial Respiratory rate (rpm)** | | | |
| Mean (SD) | 20.7 (6.30) | 18.7 (4.45) | 20.5 (6.21) |
| Median [Min, Max] | 18.0 [0, 99.0] | 18.0 [0, 96.0] | 18.0 [0, 99.0] |
| Missing | 1820 (1.9%) | 188 (2.6%) | 2008 (2.0%) |
| **Initial Systolic Blood Pressure (mmHg)** | | | |
| Mean (SD) | 143 (28.3) | 143 (24.4) | 143 (28.1) |
| Median [Min, Max] | 142 [0, 265] | 140 [1.00, 288] | 142 [0, 288] |
| Missing | 2991 (3.2%) | 388 (5.4%) | 3379 (3.3%) |
| **Initial Diastolic Blood Pressure (mmHg)** | | | |
| Mean (SD) | 82.9 (17.7) | 86.4 (15.6) | 83.2 (17.6) |
| Median [Min, Max] | 83.0 [0, 200] | 86.0 [4.00, 182] | 83.0 [0, 200] |
| Missing | 3114 (3.3%) | 397 (5.5%) | 3511 (3.5%) |
| **Initial Oxygen saturations (%)** | | | |
| Mean (SD) | 95.3 (5.41) | 97.1 (2.84) | 95.4 (5.29) |
| Median [Min, Max] | 97.0 [11.0, 100] | 98.0 [18.0, 100] | 97.0 [11.0, 100] |
| Missing | 2543 (2.7%) | 329 (4.6%) | 2872 (2.8%) |
| **Initial temperature (Celsius)** | | | |
| Mean (SD) | 37.0 (0.965) | 36.8 (0.735) | 37.0 (0.952) |
| Median [Min, Max] | 36.9 [31.7, 42.1] | 36.8 [33.0, 40.7] | 36.9 [31.7, 42.1] |
| Missing | 5935 (6.3%) | 796 (11.0%) | 6731 (6.6%) |
| **Initial Pain Score** | | | |

(*Continued*)

**Table 1.** (Continued)

| | Unavoidable (N = 94294) | Avoidable (N = 7228) | Overall (N = 101522) |
|---|---|---|---|
| Mean (SD) | 3.10 (3.58) | 2.94 (3.50) | 3.09 (3.57) |
| Median [Min, Max] | 1.00 [0, 10.0] | 0 [0, 10.0] | 1.00 [0, 10.0] |
| Missing | 26475 (28.1%) | 2073 (28.7%) | 28548 (28.1%) |
| **Self Mobile** | | | |
| Yes | 26079 (87%) | 3792 (13%) | 29871 |
| No | 68215 (95%) | 3436 (5%) | 71651 |
| **Initial NEWS2 score** | | | |
| 0 | 20807 (90%) | 2194 (10%) | 23001 |
| 1 | 16801 (90%) | 1779 (10%) | 18580 |
| 2 | 10527 (92%) | 928 (8%) | 11455 |
| 3 | 9910 (94%) | 610 (6%) | 10520 |
| 4 | 6899 (95%) | 330 (5%) | 7229 |
| 5 | 5172 (97%) | 186 (3%) | 5358 |
| 6 | 4564 (97%) | 128 (3%) | 4692 |
| 7 | 3313 (98%) | 60 (2%) | 3373 |
| 8 | 2696 (99%) | 40 (1%) | 2736 |
| 9 | 2033 (99%) | 16 (1%) | 2049 |
| 10 | 1475 (99%) | 10 (1%) | 1485 |
| 11 | 969 (99%) | 9 (1%) | 978 |
| 12 | 595 (100%) | 2 (0%) | 597 |
| 13 | 441 (100%) | 2 (0%) | 443 |
| 14 | 241 (99%) | 3 (1%) | 244 |
| 15 | 138 (100%) | 0 (0%) | 138 |
| 16 | 58 (100%) | 0 (0%) | 58 |
| 17 | 34 (100%) | 0 (0%) | 34 |
| 18 | 13 (100%) | 0 (0%) | 13 |
| 19 | 2 (100%) | 0 (0%) | 2 |
| Missing | 7606 (89%) | 931 (11%) | 8537 |

can be perceived as less optimistic from the outset [27]. Table 2 is a brief summary of the performance measures being used to evaluate the model.

**Calibration.** Calibration was assessed using Spiegelhalter's Z-test and calculated using the Rmisc package v1.5 [28]. The interpretation of this Z-test is such that a statistically significant

**Table 2. Model performance measures.**

| Test | Description | Statistic | Interpretation |
|---|---|---|---|
| Calibration | Assessment of whether the predicted probabilities match with the observed probabilities. | O:E ratio | A perfect O:E ratio would be 1. If the model is over-triaging, the O:E ratio will be greater than 1 as it would predict more than observed, and vice versa. |
| | | Spiegelhalter's z-test | A Spiegelhalter's z-test that falls outside the interval of -1.96–1.96 will have a p-value greater than 0.05 and it means the model is miscalibrated. |
| Discrimination | Discrimination is assessing whether the model can take two random instances (one with and without the outcome) and tell them apart. | C-statistic | The C-statistic of 0.5 means the model is no better than chance at telling apart the two random instances. A C-statistic of 1 means the model will tell the two random instances apart every time. A good C-statistic achieved in prior studies for this clinical problem is 0.8. |

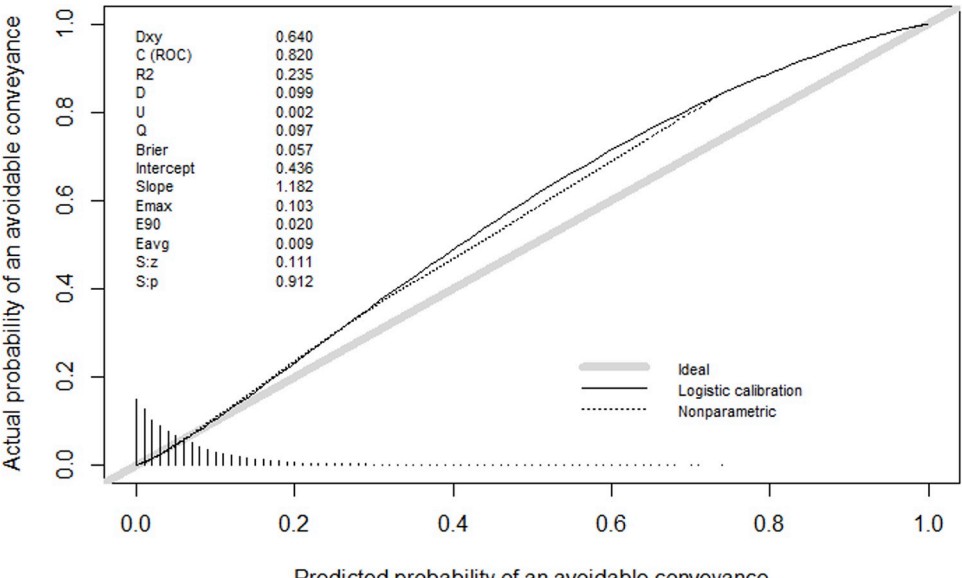

**Fig 1. Full model calibration plot.**

test result means the model is miscalibrated as the null hypothesis is a well calibrated model. The initial model was miscalibrated with a Spiegelhalter's Z-test of -3.668 (p = 0.001). Therefore, the weighting of the positive class was tuned to two decimal places to yield the smallest Z-test with no statistical significance. The optimum value for scale_pos_weight was 0.95 which gave a Spiegelhalter's Z-test of 0.111 (p = 0.912). The ratio of the observed and expected (O:E) was 1.042 (95% CI 1.02–1.07). The full calibration plot with intercept and slope can be found in Fig 1.

**Discrimination.** The C-statistic for the full model was 0.82 (95% CI 0.815–0.824). The optimum cut point was 0.121, which gave a specificity of 0.87 and a sensitivity of 0.54. The ROC curve with different thresholds including the optimal threshold (marked with a star) can be found in Fig 2. The threshold was chosen as the 'closest top left' point mathematically. Experiments were performed by maximising specificity, but the model was unstable, and the sensitivity decreased by such a significant amount that it would miss-classify far more often than it would classify.

Using the optimal cut point, the full model had an accuracy of 0.85 (95% CI 0.847–0.852). The model had a preference towards specificity as it was predicting health and not disease. The positive predictive value (PPV) was 0.25 (95% CI 0.24–0.25) and the negative predictive value was 0.96 (95% CI 0.96–0.963).

**Model updating.** The meta-analysis was undertaken using the framework by Debray et al. and used the metamisc package v0.2.5 [29, 30]. In the meta-analysis of clusters, the C-statistic was found to be 0.81 (95%CI 0.79–0.83). The prediction interval was between 0.73 and 0.87. Fig 3 shows the forest plot of C-statistic results for each cluster. The hyperparameters of each model can be found in S4 Appendix. The meta-analysed O:E ratio was 0.995 (95% CI 0.97–1.03) with a prediction interval between 0.93 and 1.06. In S5 Appendix, there are calibration plots and ROC curves for each model developed.

**Fair machine learning analysis.** In the analysis of fair machine learning, each demographic was assessed on two criteria. The first was comparing the probability distribution of each category within the variable and the second was examining how many were misclassified in each category. If age is left in as a candidate variable, the model becomes more accurate but

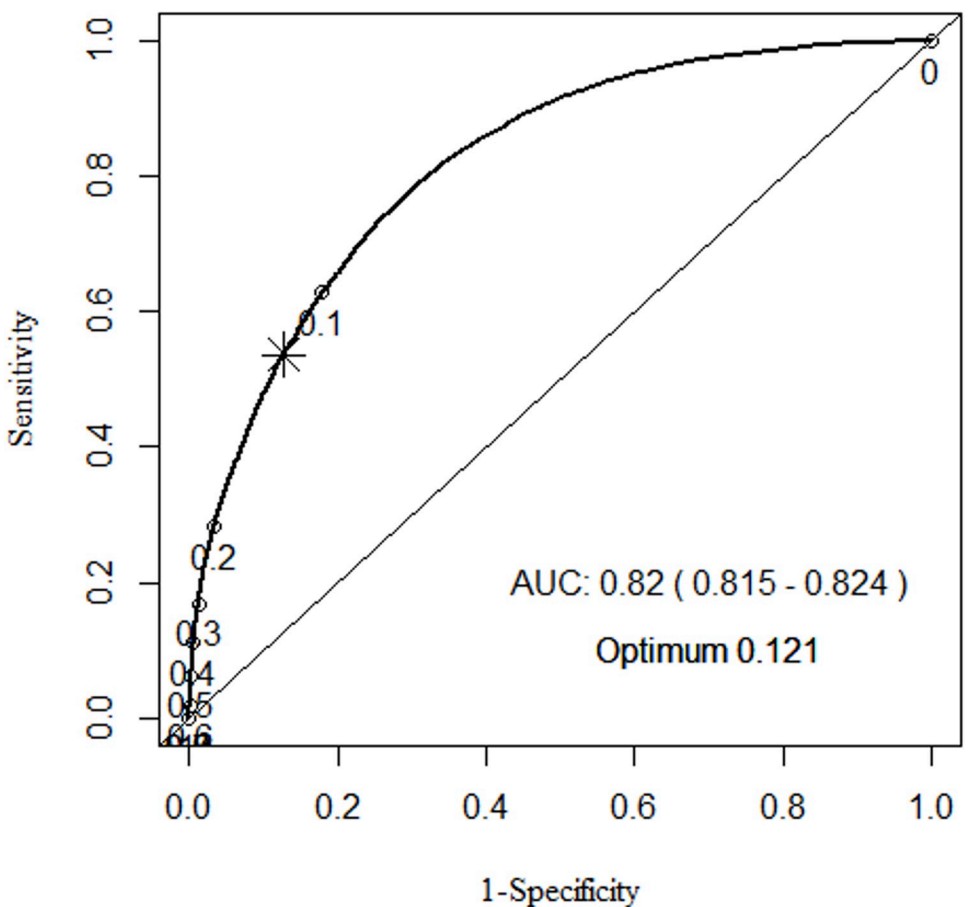

**Fig 2. ROC curve of the full model.**

introduces a bias towards younger patients. When excluded, the model slightly decreases in performance but removes the bias. There were no significant differences in the mean probabilities, distributions, or misclassification for any of the demographic variables assessed. This included ethnicity, gender, and social deprivation. More information can be found in S6 Appendix.

**Misclassification analysis.** There were 3880 (3.8%) true positive predictions where the model correctly identified an avoidable ambulance conveyance and 82,340 (81.1%) true negatives where it identified an unavoidable conveyance. There were 11,954 (11.8%) false positives and 3348 (3.3%) false negatives. This gave a misclassification rate of (0.151).

**Variable importance.** Variable importance can be broken down into three features—frequency (weight), coverage and gain. Frequency represents how many times a particular feature appears in the trees of the full model as a percentage of all the frequencies. Coverage is the number of instances that are contained within a feature when it is used as a split. Gain is the relative contribution of each feature to the whole model. Figs 4–6 show the frequency, coverage and gain for the model.

## Discussion

This study used a large sample of conveyed ambulance patients linked to their ED record to derive a clinical decision support model. Two different systematic reviews concluded that the

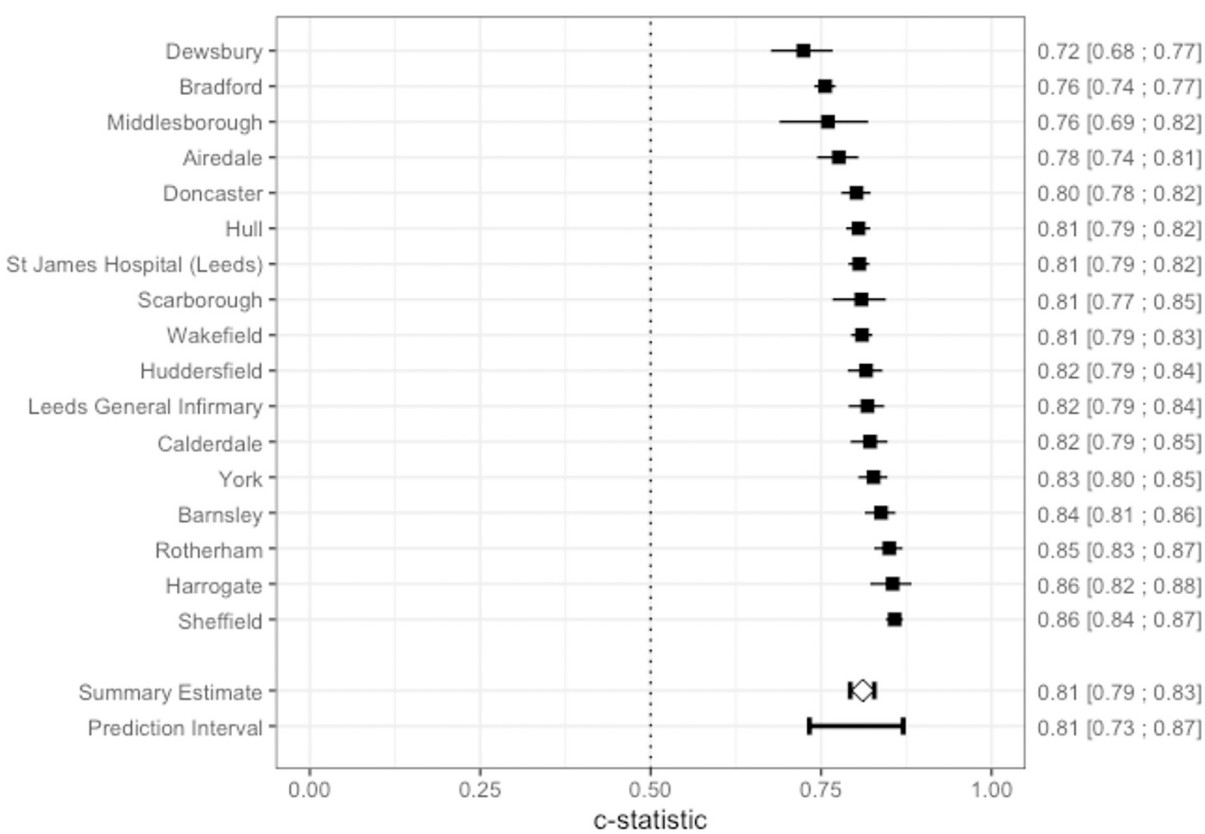

**Fig 3. Meta-analysis of cluster discrimination.**

most effective clinical decision support should be computer-based, providing support as part of the natural workflow, offering practical advice and being available at the time of decision making. Computerised Clinical Decision Support (CCDS) in the prehospital system increasingly plays an important role in delivering efficient care that can meet the needs of its users. In an environment where information is difficult to obtain but decisions are crucial and time limited, CCDS tools appear to offer a potential solution. In a Department for Health and Social Care review of operational productivity of ambulance services in England, the first recommendation for future contracting was for ambulance services to have 'technology, processes and systems in place to support clinical decision making' [31].

Computerised decision support is relatively novel to clinicians on scene. This is owing to the requirement of electronic patient care records. In Yorkshire Ambulance Service, ePCRs were only fully launched in July 2019, and this formed a barrier to data availability. However, evidence is mounting about the benefits of on-scene CCDS, and the results in this study could have the greatest benefit if a prospective tool is used on scene with the patient.

One of the more neoteric advancements of on scene CCDS is predicting end diagnosis to expedite specialist care or to instigate earlier treatment. As an example, The Japanese Urgent Stroke Triage Score using Machine Learning (JUST-ML) predicted a major neurological event such as a large vessel occlusion, subarachnoid haemorrhage, intracranial haemorrhage or cerebral infarction better than any other available model [32]. The benefit of predicting a major neurological event in the pre-hospital phase of care is that it can steer transport destination decisions to ensure the right patients go to a stroke unit for specialist care. Predicting a downstream outcome has been seen in many clinical conditions including Acute Coronary

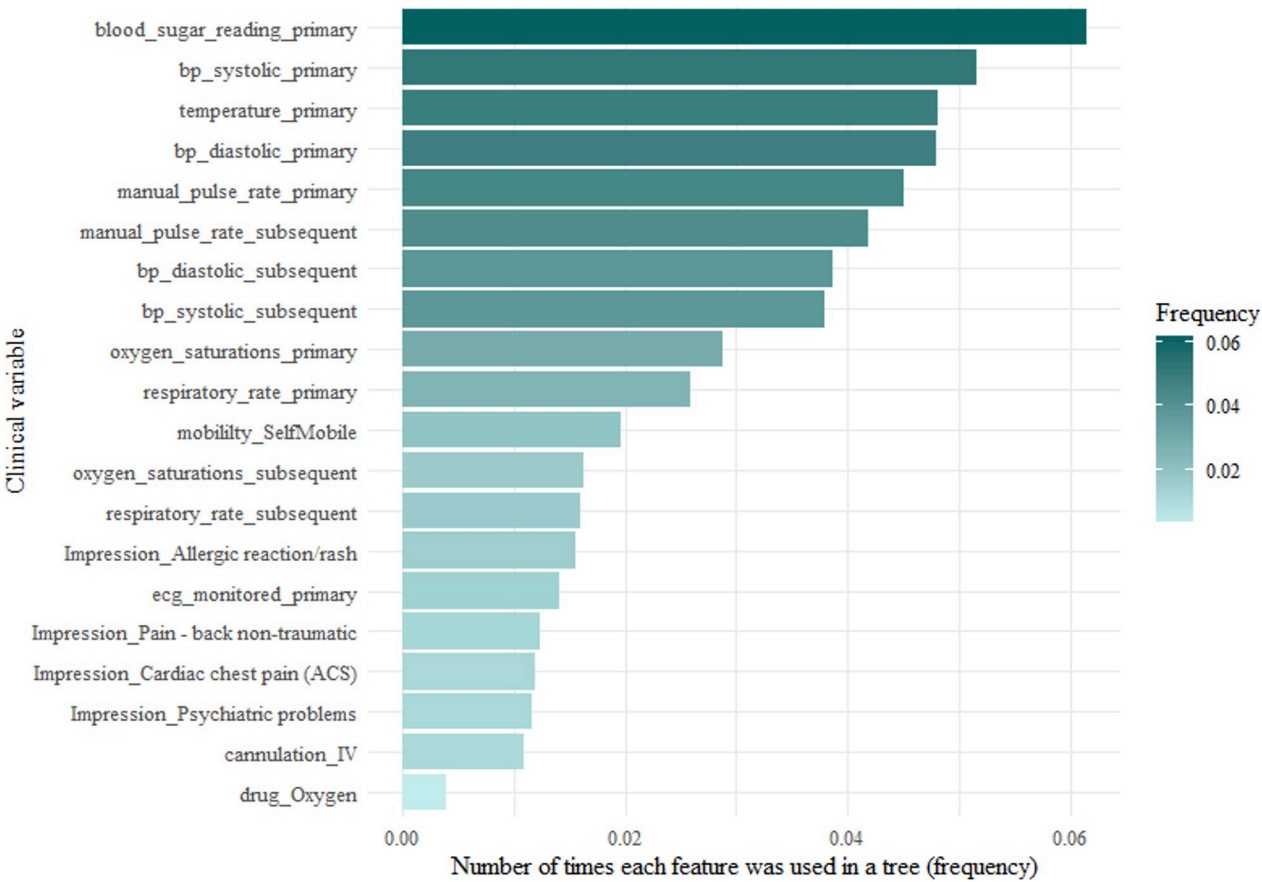

**Fig 4. Top 20 variables used in the full model by frequency.**

Syndrome (ACS) and major trauma [33–35]. The results of this study cannot extend to predicting an end diagnosis, however, they support the idea of modifying a care plan according to the outcome of a CCDS tool. The model has demonstrated it can predict avoidable ambulance conveyances and contributes evidence that computerised decision support can not only predict a high acuity outcome, but also low.

In the SAFER1 trial, the computerised decision support tool was embedded into the ePCR [36]. In the qualitative evaluation, it was found that the paramedics who had access to the tool were twice more likely to refer patients to a falls service than those without. However, the paramedics only applied the tool in 12% of eligible patients. One of the barriers to implementation identified in the qualitative element to the study included the labour involved in accessing and using the tool. This resonates with the work of Kawamoto et al [37]. In their systematic review, they were aiming to identify key features of success in the implementation of clinical decision support systems. The most important feature was automation and ensuring that the effort on the end user was minimised. The reason that machine learning algorithms were considered for developing the SINEPOST model was their potential accuracy and ability to be embedded in an electronic healthcare system. Whilst the Occam's razor approach of making the model as simple as possible was the intended philosophy of the SINEPOST model, machine learning algorithms can be complicated, if needed, and still provide automated prediction.

Decision support systems that are already in place for triaging patients include the paramedic pathfinder and the Manchester Triage System (MTS) [16, 17, 38]. The outcomes of

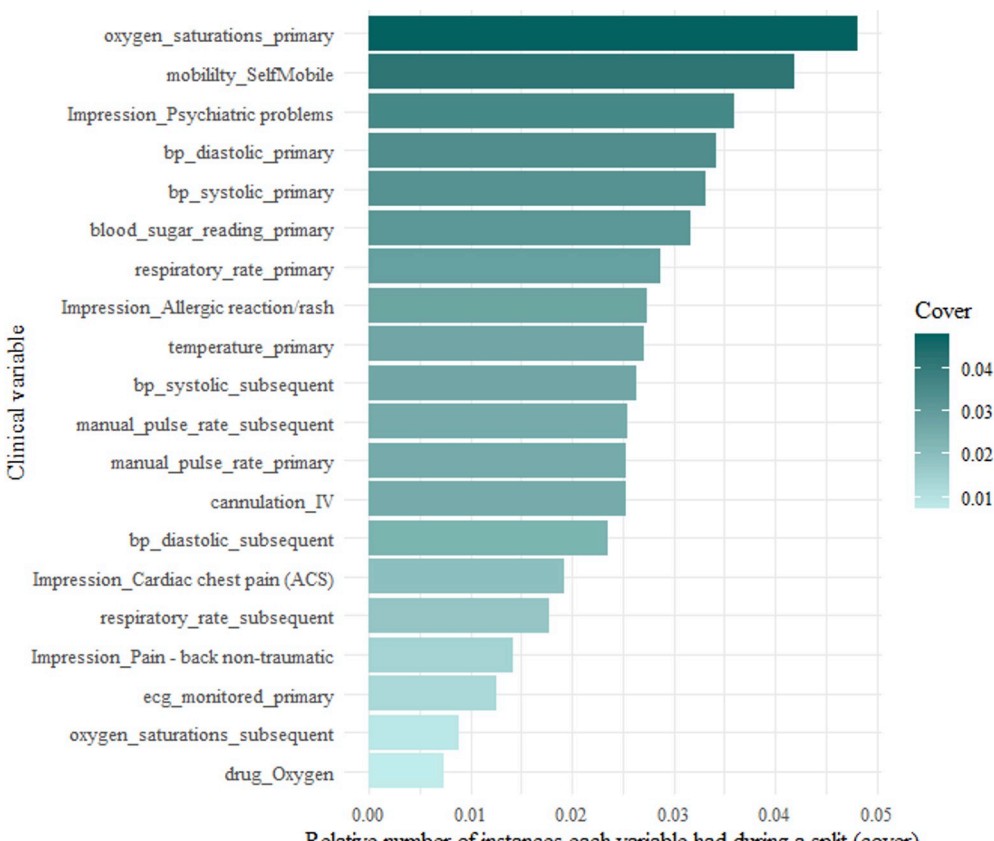

**Fig 5. Top 20 variables with the greatest number of instances when splitting (cover).**

these tools are different, and so it would be inappropriate to compare performance between them. The intended use of these tools was to risk stratify patients to support non conveyance decisions.

This study could have adopted a different strategy, taking a non-conveyed sample and a conveyed sample to create a prediction model predicting non-conveyance. However, the gold standard used would be paramedic decision making, and therefore the model would only be as good as what is already out there. This is a limitation in both the paramedic pathfinder and the MTS. The strength in this study was taking information that the ambulance crew would not know and predicting that information for them to use whilst they were on scene. The results of this study have demonstrated that using the prehospital variables, it is entirely possible to predict the experience they may have if they were transported to ED. This brings with it a benefit to paramedic decision making. One Canadian study demonstrated it was feasible to use a computer algorithm to redirect nonemergent patients away from the ED towards sub-acute centres such as walk-in centres. This had both system and patient benefits (such as satisfaction) [39].

In the study by Miles et al. they explored paramedic decision making using a mixed methodology [15]. In the qualitative part, it was found that paramedics either framed a decision around the scene, or the ED. When they framed the decision around the scene, their language would often be why it is not safe to be left at home, or that the patient requires a GP appointment (for example). When it was framed around the ED, the justifications would be anchored to the patient either receiving a certain benefit from attending, or that the ED would probably not find anything abnormal [15]. The findings from this study have the opportunity to support

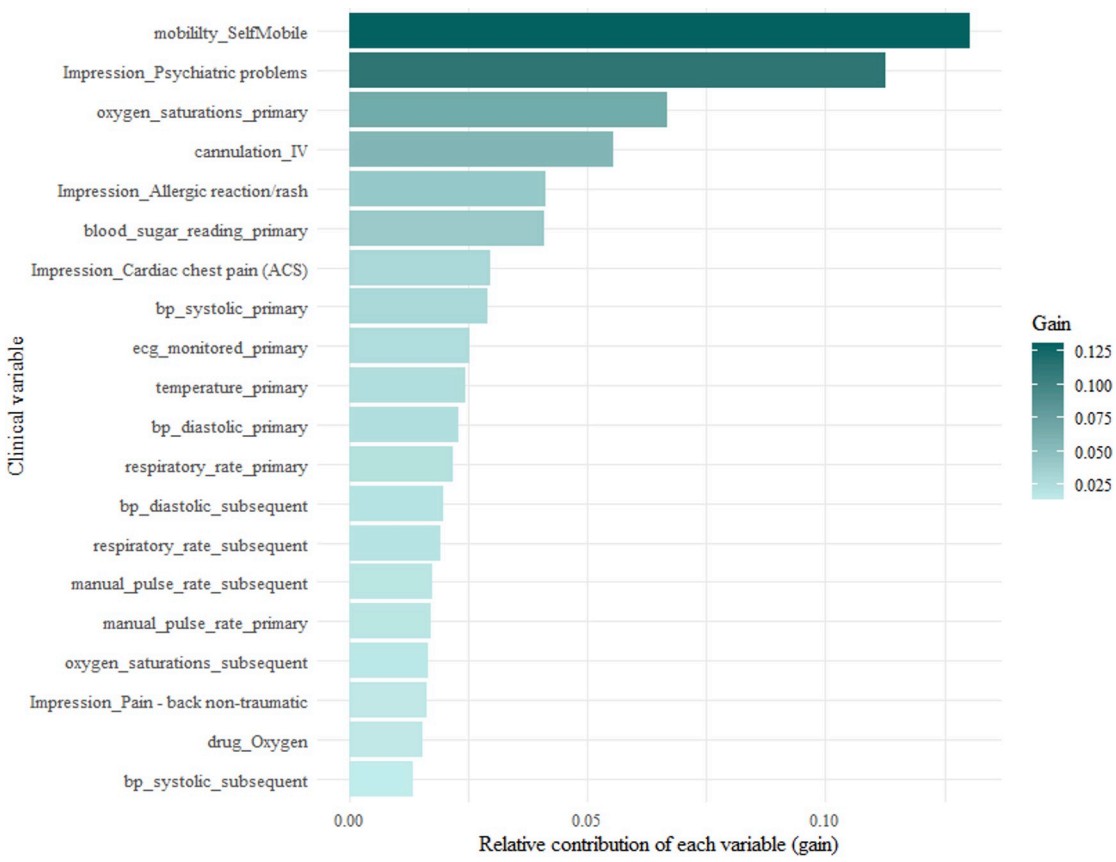

**Fig 6. Top 20 variables with the highest relative contribution to the full model (gain).**

those who use the ED to frame their decisions. By knowing what the predicted probability is, it provides new information to them that would not have been available for decision making. However, perhaps the largest benefit to transport decision making on scene from this study is the revealing of clinically important variables that should be accounted for in making such a decision.

## Feature importance

A unique and novel finding from this study was the identification of six clinical impressions that were important in predicting avoidable conveyances. There were six clinical impressions that featured in the top twenty. The most important was patients presenting with psychiatric problems. This could be a reflection on the experience of mental health presentations at the ED. They rarely require investigations or treatments that physical health presentations may require. The main purpose of the ED for these patients is to offer a place of safety and access to a mental health practitioner who can better meet their care need. Other clinical impressions were allergic reactions, cardiac chest pain, head injury, non-traumatic back pain, minor cuts, and bruising. These have been previously identified in observational studies as being associated with a non-urgent ambulance conveyance [10, 40]. All physiological observations appeared in the top twenty, however the NEWS2 score did not. Only three NEWS2 scores were included in the full model. A NEWS2 score of 0 appeared as the 31st variable, a score of 1 as the 58th, a score of 2 as the 78th and a score of 5 as the 93rd. This may mean that low NEWS2

scores are not strong predictors of an avoidable conveyance attendance. This is an interesting finding, as the decision tree should have associated higher NEWS2 scores with information gain of ruling out an avoidable conveyance. Conversely, it omitted most NEWS2 scores during recursive feature elimination. In the full model and all the clusters, the frequency, cover and gain did not change rank order, which shows the stability of their importance. When predicting high acuity, it is often easier to find significant variables as physiological observations such as pulse rate and respiration rate will change when patients are acutely unwell. However, when predicting avoidable conveyances, physiological observations will often be normal. Interestingly, there were clinical variables more important than physiological observations, which have featured in other triage models as main candidate predictors [41, 42]. In the development of decision tree models, splits are made based on the information gained. This can be either gain in deciding what an avoidable conveyance patient is or gain in deciding what an avoidable conveyance patient is not. As such, variables associated with higher acuity appear high in variable importance as they rule out necessary attendances. The algorithm has identified signals of higher acuity patients with high prevalence of completion within the ePCR. For example, delivering advanced life support to someone in cardiac arrest does not often happen in the overall case-mix of ambulance patients. Therefore, the skills and procedures associated with undertaking ALS were rarely captured and were not identified as important. However, far more patients had the clinical procedure of intravenous cannulation or monitored by ECG, and it appeared as the fifth and eighth most important variables. This theory can be extended to the patient's mobility. In the model, a patient's mobility status is important, as being stretcher bound, self-mobile or needing a carry chair all featured in the top twenty.

## Model performance

The model was well calibrated with a meta-analysed O:E ratio of 0.99 (95% CI 0.96–1.02). This means that the model is making accurate predictions across all values. The model is also successful in distinguishing between an avoidable ambulance conveyance and one that needed transport to hospital with a C-statistic of 0.81 (95% CI 0.79–0.83). The optimal threshold for classification was 0.125 which appears low, but so is the proportion of avoidable ambulance conveyances and this reflects the class imbalance. The model provided many false negatives with a sensitivity of 0.58, meaning that 42% of patients who were classified as needing ED care were avoidable conveyances. The choice of threshold is a point of discussion. It could be adjusted to a higher or lower value, but this would influence the sensitivity and specificity. To illustrate, the ROC curve in Fig 2 shows the thresholds above 0.2 have limited effect on the specificity but a large effect on sensitivity. If the threshold was changed to 0.2 for example, the sensitivity drops dramatically to 0.28. The optimum threshold was chosen to be the highest specificity with the highest sensitivity. Also known as a balanced approach. It was also possible to take the Youden index, which would place the threshold at the nearest point to the top left corner, but this placed too much of a penalty on specificity to create a functioning tool.

The meta-analysis of clusters revealed that there were no significant performance differences between test sets in urban areas, rural areas, or coastal areas. There were significant differences in the calibration slopes as seen in S7 Appendix; however, this was at the latter part of the plots where predicted outcome was rare. They all produced O:E ratios that were acceptable except for two smaller test sets (Dewsbury hospital and James Cook University Hospital) who had significant under-triage. Sheffield, Leeds, York, and Hull are all large teaching hospitals, and as illustrated by Fig 3, there were no significant differences between these. Furthermore, there was no significant difference between the large teaching hospitals and the smaller district general hospitals.

There was only a prevalence of 7% for avoidable conveyance attendances in the study sample. This is fewer cases than the literature had previously reported (9–13%) [7, 10–12]. This may appear low; however, the quantity of high acuity patients is similar, indicating that to predict avoidable conveyance and high-acuity would be predicting the two tails of a normal distribution. Future studies should examine the mid-acuity patients and begin to unpick differences between these patients to improve on the outcome definition of a patient who is unlikely to gain a clinical benefit from being transported to a higher-acuity clinical setting than community care.

## Limitations

This study has its limitations. It was a retrospective, observational study using routine data. A strength of using routine data is the ability to use large volumes of patient episodes, which can produce accurate models. A limitation, however, is that it is not feasible to tailor data collection to the project. It is only possible to use what is routinely collected, which unfortunately relinquishes any control over missing data. Another limitation is the computational expense of selecting an algorithm with many hyperparameters. It would take a significant amount of time to be able to scan all combinations of hyperparameters through a grid search every time a model was developed. As such, the grid was restricted. The anticipated impact of the restricted grid search is expected to be minimal as the differences in AUC performance (the evaluation metric of choice) had a narrow interval of between 0.7 and 0.85. The validation does not benefit from true external validation, and it would be a sensible conclusion to revisit the definition of an avoidable ambulance conveyance, or indeed the taxonomy of how prehospital care systems classify their patients based on their need before further validation of the SINEPOST model.

## Interpretation

This study can conclude that it is possible, with good accuracy to predict an avoidable ambulance conveyance to the ED using prehospital clinical data. The XGBoost model developed here, known as the SINEPOST model, can discriminate between those with non-urgent needs and those without. It can also accurately provide what the probability of an avoidable conveyance is. The model does not bias different ages, ethnicities, genders, or Indices of Deprivation. It is robust to all different prehospital settings. If this Fig was applied to national level data in England, the predictive model could support 85,560 conveyance decisions per month to change to non-conveyance. This is based on the latest NHS England Ambulance Quality Indicators which identified 372,002 ambulance transports to the ED in November 2021 [43]. However, to maximise its potential if it was to be transformed into a computerised clinical decision support tool; there needs to be a more robust definition of what an avoidable conveyance should be. It is recommended to revise the taxonomy of prehospital patients according to the care setting they need, as opposed to the paradigm of describing patient acuity. This has shown success in Canada already, with a computer algorithm demonstrating it is possible to redirect nonemergent patients away from the ED towards sub-acute centres such as walk-in centres. This had both system and patient benefits (such as patient satisfaction) [39].

It would also be beneficial to undertake studies into the risk tolerance of policy makers, ambulance services and the public when it comes to transporting low- or mid-acuity patients to the ED.

## Supporting information

**S1 Appendix. Data flow diagram.**
(PDF)

**S2 Appendix. List of Emergency Departments included in this study.**
(PDF)

**S3 Appendix. Included variables in the model.**
(PDF)

**S4 Appendix. Hyperparameter values per cluster.**
(PDF)

**S5 Appendix. ROC and calibration curves for the IECV models.**
(PDF)

**S6 Appendix. Fair machine learning analysis.**
(PDF)

**S7 Appendix. All candidate variables from the ePCR dataset.**
(PDF)

## Author Contributions

**Conceptualization:** Jamie Miles, Janette Turner, Suzanne Mason.

**Data curation:** Richard Campbell.

**Formal analysis:** Jamie Miles, Richard Jacques.

**Funding acquisition:** Jamie Miles, Suzanne Mason.

**Investigation:** Jamie Miles.

**Methodology:** Jamie Miles, Richard Jacques.

**Project administration:** Jamie Miles, Richard Campbell.

**Software:** Jamie Miles.

**Supervision:** Richard Jacques, Janette Turner, Suzanne Mason.

**Validation:** Jamie Miles.

**Visualization:** Jamie Miles.

**Writing – original draft:** Jamie Miles.

**Writing – review & editing:** Jamie Miles, Richard Jacques, Richard Campbell, Janette Turner, Suzanne Mason.

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
