## [Decision Letter · Decision Letter 0]

27 Jul 2022

PONE-D-22-17407The Safety INdEx of Prehospital On Scene Triage (SINEPOST) study: the development and validation of a risk prediction model to support ambulance clinical transport decisions on-scene.PLOS ONE

Dear Dr. Miles,

Thank you for submitting your manuscript to PLOS ONE. After careful consideration, we feel that it has merit but does not fully meet PLOS ONE’s publication criteria as it currently stands. Therefore, we invite you to submit a revised version of the manuscript that addresses the points raised during the review process.

We look forward to receiving your revised manuscript.

Kind regards,

Yong-Hong Kuo

Academic Editor

PLOS ONE

Journal Requirements:

4. Please ensure that you refer to Figures 2 and 3 in your text as, if accepted, production will need this reference to link the reader to the figure.

5. Please upload a copy of Figures 13 and 14, to which you refer in your text on pages 26 and 20. If the figure is no longer to be included as part of the submission please remove all reference to it within the text.

Additional Editor Comments:

Both referees believe that the manuscript has good potential to be published at PLOS ONE. While both recommend "minor revision", some comments from the referees may need quite some efforts to address them. I recommend "major revision".

Reviewers' comments:

Reviewer's Responses to Questions

**Comments to the Author**

1. Is the manuscript technically sound, and do the data support the conclusions?

Reviewer #1: Yes

Reviewer #2: Yes

2. Has the statistical analysis been performed appropriately and rigorously? 

Reviewer #1: I Don't Know

Reviewer #2: Yes

3. Have the authors made all data underlying the findings in their manuscript fully available?

Reviewer #1: No

Reviewer #2: Yes

4. Is the manuscript presented in an intelligible fashion and written in standard English?

Reviewer #1: Yes

Reviewer #2: No

5. Review Comments to the Author

Reviewer #1: Thank you for the opportunity to review the paper titled “The Safety INdEx of Prehospital On Scene Triage (SINEPOST) study: the development and validation of a risk prediction model to support ambulance clinical transport decisions on-scene”. The objective of this study was to determine whether prehospital information can predict avoidable attendance. This is likely one of the most pressing questions facing paramedic services today as demand exceeds capacity. There are few decision support tools that exist at present to help with determining whether a case is an avoidable attendance. Paramedics can have challenges determining medical necessity and need for hospitalization.

As you will see, I have made several minor suggestions for how the paper might be improved.

Background:

• Here is another recent reference on the potential for harm due to offload delay.

o Dawson LP et al. Med J Aust. 2022 Jun 23. doi: 10.5694/mja2.51613. Online ahead of print. The influence of ambulance offload time on 30-day risks of death and re-presentation for patients with chest pain

• Here are a couple of comments for consideration: 1) It might also be helpful to situate this paper in the context of systems where paramedics transport most patients to the ED (and may not be making the decision) unless the patient refuses transport. I think this work could inform those systems in developing clinical pathways based on the tool developed here. 2) I’m also wondering what an appropriate over-triage might be? 9-32% over triage may not be that bad.

Methods

• Please define HES A&E and ECDS were defined.

• Participants - should this state “face to face paramedic contact from Yorkshire ambulance service”?

• More info on the scaling of the NEWS2 would be helpful. How are the NEWS2 categories determined?

• How many clinical interventions were considered?

• Patient’s mobility is interesting. Please clarify – this is really about how the patient was extricated from the scene and not about their day-to-day level of mobility and balance. Is this a drop-down selection in the ePCR system?

• Were any indicators of frailty considered?

• Please clarify what is meant by social variables and network variables. What types of data do paramedics routinely document here?

• Sample size – should the conservative estimate state 0.09? Please check if this is the correct reference and prevalence.

• This is a great description of the data linkage process.

Results

• Are you able to report on how many records were successfully linked by NHS Digital?

• Conveyance is repeated in the first paragraph of results.

• Were the first set of vital signs used prior to interventions?

• Please check the discrimination section. Should it be 0.820 (95% CI 0.815-0.824)?

• Typo in model updating section – should be fig 3 instead of fig 14

• Figures 4-6 were difficult to read. Consider removing and describe.

Discussion

• Does the model better predict when to transport to a tertiary care versus a smaller community hospital?

• There is reference to a figure 13 which does not exist.

• I think more thought should be placed on how to situate this work within the academic literature.

o This study may be helpful for determining what additional patients may be suitable for non-conveyance.

Emergency department interventions that could be conducted in subacute care settings for patients with nonemergent conditions transported by paramedics: a modified Delphi study Ryan P Strum 1, Walter Tavares, Andrew Worster 2, Lauren E Griffith 2, Andrew P Costa 2CMAJ Open 2022 Jan 11;10(1):E1-E7. doi: 10.9778/cmajo.20210148. Print Jan-Mar 2022

• Overall, I thought this was an interesting paper and moves the field forward towards understanding how paramedics can make better decisions regarding transport dispositions. There may be opportunities to be more concise throughout.

Reviewer #2: The proposed manuscript (ms) greatly contributes to the sciejce. I have not found fundamental issues making the proposed ms rejected. Despite some short comings,the ms is relatively technically sound, presented in an intelligible fashion and not written in good quality english. Appropriately chosen terms and abbreviations are clearly explained and used for better understanding. Impact of the study is well highlighted. I am suggesting a minor revision without which the proposed ms can not be accepted.

6. PLOS authors have the option to publish the peer review history of their article (what does this mean?). If published, this will include your full peer review and any attached files.

Reviewer #1: **Yes: **Judah Goldstein

Reviewer #2: No

---

## [Author Response · Author response to Decision Letter 0]

16 Sep 2022

Editor comments

This manuscript has undergone extensive reformatting to comply with PLOS ONE’s style requirements. Including those for naming files and referencing.

Thank you very much for the clarification on the journal requirements. The following section has been added:

Data availability statement

This study used patient sensitive linked data from two sources. Yorkshire Ambulance Service NHS Trust, and NHS Digital. The latter source collects routine healthcare data from all hospital trusts in England and compiles them into data products. The data products used in this study were the Emergency Care Data Set (ECDS) and the Hospital Episode Statistics Accident and Emergency (HES AE). More information on these data products can be found here: https://digital.nhs.uk/services/data-access-request-service-dars/dars-products-and-services. The minimum data set for this study is unavailable because access to both sources of data was through two data sharing agreements to process identifiable information, link the records and to analyse the data in a specific way. The data sharing agreements extended to state that the data must be destroyed once all analysis had been completed on the data. For reproducing the research, the parameters of records for Yorkshire Ambulance Service NHS Trust and NHS Digital have been outlined in the above sections. Appendix S1 in the supporting information shows a detailed flow diagram of the data processing and linking. 

The following section has been added: 

This study underwent extensive ethical review. It was first reviewed and approved by the South Yorkshire NHS Research Ethics Committee (REC) on the 20th December 2019. It was also reviewed and approved by the NHS Confidentiality Advisory Group (CAG) on the 14th July 2020. During the data sharing agreement stage, it was further reviewed and approved by the NHS Digital Independent Group Advising on the Release of Data (IGARD) team on the 15th Feb 2021. This study used patient data without written or verbal patient consent as it was not feasible to achieve this with the large volume of retrospective data. To mitigate this, the patient identifiers were first screened against the NHS National data opt out. This removed all patient episodes where the patient had previously stated they did not want their data used for the purposes of research. To further mitigate this, privacy notices were shared on both the Yorkshire Ambulance Service NHS Trust and the University of Sheffield websites. These contained contact details to remove participants from the study, prior to pseudonymisation. 

4. Please ensure that you refer to Figures 2 and 3 in your text as, if accepted, production will need this reference to link the reader to the figure.

Figures updated

5. Please upload a copy of Figures 13 and 14, to which you refer in your text on pages 26 and 20. If the figure is no longer to be included as part of the submission please remove all reference to it within the text.

Noted and removed.

References changed in the manuscript, and a section added at the end to index the supporting information. All supporting information now individually saved and converted to .pdf.

Reviewer 1 comments

Reviewer #1: Thank you for the opportunity to review the paper titled “The Safety INdEx of Prehospital On Scene Triage (SINEPOST) study: the development and validation of a risk prediction model to support ambulance clinical transport decisions on-scene”. The objective of this study was to determine whether prehospital information can predict avoidable attendance. This is likely one of the most pressing questions facing paramedic services today as demand exceeds capacity. There are few decision support tools that exist at present to help with determining whether a case is an avoidable attendance. Paramedics can have challenges determining medical necessity and need for hospitalization.

As you will see, I have made several minor suggestions for how the paper might be improved.

Background:

• Here is another recent reference on the potential for harm due to offload delay.

o Dawson LP et al. Med J Aust. 2022 Jun 23. doi: 10.5694/mja2.51613. Online ahead of print. The influence of ambulance offload time on 30-day risks of death and re-presentation for patients with chest pain

Thank you very much for highlighting this new study. The following text has been added into the background:

“Studies have been more specific in identifying harm that has occurred with certain diseases. It has been shown that delayed handover in patients with non-traumatic chest pain is associated with a greater risk of 30-day mortality”

• Here are a couple of comments for consideration: 1) It might also be helpful to situate this paper in the context of systems where paramedics transport most patients to the ED (and may not be making the decision) unless the patient refuses transport. I think this work could inform those systems in developing clinical pathways based on the tool developed here. 

Introduction text added: 

“It is recognised that in some systems transport decisions are not clinician-made and patient-centred, but financially driven through payment policies.[13,14] However, these policies are beginning to adapt to the modern case-mix of patients and as such, the adoption of a transport decision support tool would be of high benefit and importance.”

2) I’m also wondering what an appropriate over-triage might be? 9-32% over triage may not be that bad.

I think this is a very interesting point that you have raised. It would be beneficial to understand what risk tolerance policy makers, ambulance services and the public would have in over triage. It does fall outside of the remit of this study; however, I have added the following into the interpretation section:

“It would also be beneficial to undertake studies into the risk tolerance of policy makers, ambulance services and the public when it comes to transporting low- or mid-acuity patients to the ED.”

Methods

• Please define HES A&E and ECDS were defined.

Both are now defined.

• Participants - should this state “face to face paramedic contact from Yorkshire ambulance service”?

Yes, thank you for spotting this. The text has been changed. 

• More info on the scaling of the NEWS2 would be helpful. How are the NEWS2 categories determined?

The following text has been added, with a reference to the Royal College of Physicians document.

“The NEWS2 assigns points between 0 and 3 to physiological variables depending on how deranged they are. The minimum NEWS2 score is 0, and the maximum is 20.[21]”

• How many clinical interventions were considered?

Sixteen in total. The full candidate list has now been included as appendix S7, with the following reference text at the start of the predictor section:

“Appendix S7 displays all candidate variables, example values, justification for inclusion and assigned parameters within each variable.”

• Patient’s mobility is interesting. Please clarify – this is about how the patient was extricated from the scene and not about their day-to-day level of mobility and balance. Is this a drop-down selection in the ePCR system?

Following text added for clarification:

“This variable was how the patient was able to move to and from the ambulance and was a categorical variable.”

• Were any indicators of frailty considered?

Unfortunately not. I think this is an important candidate variable to consider, and the ambulance services are starting to collect this in their ePCRs. However, at the time of model development, it was unavailable. 

• Please clarify what is meant by social variables and network variables. What types of data do paramedics routinely document here?

Section clarified to the following:

“Social variables were included as binary variables. These were included as surrogates to determine the level of external support the patient has. These include variables such as GP details recorded, social worker recorded etc. It also included referral variables if the patient was referred to a service such as falls, safeguarding or diabetes clinic etc.”

• Sample size – should the conservative estimate state 0.09? Please check if this is the correct reference and prevalence.

Yes, changed with thanks. Reference has been checked and they report that 8.5% of patients were non-urgent and arriving by ambulance. The sample size calculation has been updated but does not change the study validity. 

Previous studies have found a conservative estimate of the outcome prevalence to be 0.085.[12] A meta-analysis found that the average C-statistic was 0.8.[25] A preliminary analysis of a separate dataset found that there was potentially 637 parameters in the ambulance service dataset. This gave an estimated sample size of 55,676 with an anticipated 4733 event and an events per parameter (EPP) of 7.43.

• This is a great description of the data linkage process.

Results

• Are you able to report on how many records were successfully linked by NHS Digital?

Yes, the following text has been added: 

“Previous data linkage methodology with NHS Digital used an eight stage hierarchical probabilistic matching algorithm.[26] However, the ECDS data product could only be linked using the unique identifier of NHS number, which renders the linkage process to be largely deterministic. As a result, all patient records sent to NHS Digital with an NHS number were successfully linked, whereas those without an NHS number were not. This resulted in 195078 (66%) of the total cohort excluded from the analysis. A comparison of the successfully linked cohort and the unlinked cohort revealed no fundamental differences.”

• Conveyance is repeated in the first paragraph of results.

Thank you for spotting. Edited. 

• Were the first set of vital signs used prior to interventions?

This is unknown. 

• Please check the discrimination section. Should it be 0.820 (95% CI 0.815-0.824)?

Thank you very much for spotting this. Updated.

The C-statistic for the full model was 0.82 (95% CI 0.815-0.824). 

• Typo in model updating section – should be fig 3 instead of fig 14

Changed, thank you. 

• Figures 4-6 were difficult to read. Consider removing and describe.

Thank you very much for the feedback on figures 4-6. We have discussed this as a team and feel that the visual representation would be better than what would potentially be lengthy descriptions. I have replaced the figures with clearer images and hope this helps. There are elaborations into feature importance in the discussion. 

Discussion

• Does the model better predict when to transport to a tertiary care versus a smaller community hospital?

An interesting point. We did not find there was a significant difference in predictive performance between the tertiary care hospitals and more local district generals. The forest plot in figure 3 illustrates this, with the following narrative added into the discussion:

“Sheffield, Leeds, York, and Hull are all large teaching hospitals, and as illustrated by figure 3, there were no significant differences between these. Furthermore, there was no significant difference between the large teaching hospitals and the smaller district general hospitals.” 

• There is reference to a figure 13 which does not exist.

Removed, thank you.

• I think more thought should be placed on how to situate this work within the academic literature.

Thank you for this feedback, please find the additional text below:

“Two different systematic reviews concluded that the most effective clinical decision support should be computer-based, providing support as part of the natural workflow, offering practical advice and being available at the time of decision making. Computerised Clinical Decision Support (CCDS) in the prehospital system increasingly plays an important role in delivering efficient care that can meet the needs of its users. In an environment where information is difficult to obtain but decisions are crucial and time limited, CCDS tools appear to offer a potential solution. In a Department for Health and Social Care review of operational productivity of ambulance services in England, the first recommendation for future contracting was for ambulance services to have ‘technology, processes and systems in place to support clinical decision making’.58 

Computerised decision support is relatively novel to clinicians on scene. This is owing to the requirement of electronic patient care records. In Yorkshire Ambulance Service, ePCRs were only fully launched in July 2019, and this formed a barrier to data availability. However, evidence is mounting about the benefits of on-scene CCDS, and the results in this study could have the greatest benefit if a prospective tool is used on scene with the patient. 

One of the more neoteric advancements of on scene CCDS is predicting end diagnosis to expedite specialist care or to instigate earlier treatment. As an example, The Japanese Urgent Stroke Triage Score using Machine Learning (JUST-ML) predicted a major neurological event such as a large vessel occlusion, subarachnoid haemorrhage, intracranial haemorrhage or cerebral infarction better than any other available model.220 The benefit of predicting a major neurological event in the pre-hospital phase of care is that it can steer transport destination decisions to ensure the right patients go to a stroke unit for specialist care. Predicting a downstream outcome has been seen in many clinical conditions including Acute Coronary Syndrome (ACS) and major trauma.221–223 The results of this study cannot extend to predicting an end diagnosis, however, they support the idea of modifying a care plan according to the outcome of a CCDS tool. The model has demonstrated it can predict avoidable ambulance conveyances and contributes evidence that computerised decision support can not only predict a high acuity outcome, but also low. 

In the SAFER1 trial, the computerised decision support tool was embedded into the ePCR.224 In the qualitative evaluation, it was found that the paramedics who had access to the tool were twice more likely to refer patients to a falls service than those without. However, the paramedics only applied the tool in 12% of eligible patients. One of the barriers to implementation identified in the qualitative element to the study included the labour involved in accessing and using the tool. This resonates with the work of Kawamoto et al.225 In their systematic review, they were aiming to identify key features of success in the implementation of clinical decision support systems. The most important feature was automation and ensuring that the effort on the end user was minimised. The reason that machine learning algorithms were considered for developing the SINEPOST model was their potential accuracy and ability to be embedded in an electronic healthcare system. Whilst the Occam's razor approach of making the model as simple as possible was the intended philosophy of the SINEPOST model, machine learning algorithms can be complicated, if needed, and still provide automated prediction. 

Decision support systems that are already in place for triaging patients include the paramedic pathfinder and the Manchester Triage System (MTS).8,9,95 The outcomes of these tools are different, and so it would be inappropriate to compare performance between them. The intended use of these tools was to risk stratify patients to support non conveyance decisions.

This study could have adopted a different strategy, taking a non-conveyed sample and a conveyed sample to create a prediction model predicting non-conveyance. However, the gold standard used would be paramedic decision making, and therefore the model would only be as good as what is already out there. This is a limitation in both the paramedic pathfinder and the MTS. The strength in this study was taking information that the ambulance crew would not know and predicting that information for them to use whilst they were on scene. The results of this study have demonstrated that using the prehospital variables, it is entirely possible to predict the experience they may have if they were transported to ED. This brings with it a benefit to paramedic decision making. 

In the study by Miles et al. they explored paramedic decision making using a mixed methodology86. In the qualitative part, it was found that paramedics either framed a decision around the scene, or the ED. When they framed the decision around the scene, their language would often be why it is not safe to be left at home, or that the patient requires a GP appointment (for example). When it was framed around the ED, the justifications would be anchored to the patient either receiving a certain benefit from attending, or that the ED would probably not find anything abnormal.86 The findings from this study have the opportunity to support those who use the ED to frame their decisions. By knowing what the predicted probability is, it provides new information to them that would not have been available for decision making. However, perhaps the largest benefit to transport decision making on scene from this study is the revealing of clinically important variables that should be accounted for in making such a decision.”

o This study may be helpful for determining what additional patients may be suitable for non-conveyance.

Thank you for this comment. The following text has been added:

“If this figure was applied to national level data in England, the predictive model could support 85,560 conveyance decisions per month to change to non-conveyance. This is based on the latest NHS England Ambulance Quality Indicators which identified 372,002 ambulance transports to the ED in November 2021.[44]”

Emergency department interventions that could be conducted in subacute care settings for patients with nonemergent conditions transported by paramedics: a modified Delphi study Ryan P Strum 1, Walter Tavares, Andrew Worster 2, Lauren E Griffith 2, Andrew P Costa 2CMAJ Open 2022 Jan 11;10(1):E1-E7. doi: 10.9778/cmajo.20210148. Print Jan-Mar 2022

Thank you for this reference, text added into the interpretation section:

“This has shown success in Canada already, with a computer algorithm demonstrating it is possible to redirect nonemergent patients away from the ED towards sub-acute centres such as walk-in centres. This had both system and patient benefits (such as patient satisfaction).” [40]

Reviewer 2 comments

The proposed manuscript (ms) greatly contributes to the sciejce. I have not found fundamental issues making the proposed ms rejected. Despite some short comings. the ms is relatively technically sound, presented in an intelligible fashion and not written in good quality english. Appropriately chosen terms and abbreviations are clearly explained and used for better understanding. Impact of the study is well highlighted. I am suggesting a minor revision without which the proposed ms can not be accepted.

Thank you very much for the feedback.

---

## [Decision Letter · Decision Letter 1]

10 Oct 2022

The Safety INdEx of Prehospital On Scene Triage (SINEPOST) study: the development and validation of a risk prediction model to support ambulance clinical transport decisions on-scene.

PONE-D-22-17407R1

Dear Dr. Miles,

We’re pleased to inform you that your manuscript has been judged scientifically suitable for publication and will be formally accepted for publication once it meets all outstanding technical requirements.

Kind regards,

Yong-Hong Kuo

Academic Editor

PLOS ONE

Additional Editor Comments (optional):

Based on the reviewers' recommendations, I recommend Accept.

Reviewers' comments:

Reviewer's Responses to Questions

**Comments to the Author**

1. If the authors have adequately addressed your comments raised in a previous round of review and you feel that this manuscript is now acceptable for publication, you may indicate that here to bypass the “Comments to the Author” section, enter your conflict of interest statement in the “Confidential to Editor” section, and submit your "Accept" recommendation.

Reviewer #1: All comments have been addressed

Reviewer #2: All comments have been addressed

2. Is the manuscript technically sound, and do the data support the conclusions?

Reviewer #1: Yes

Reviewer #2: Yes

3. Has the statistical analysis been performed appropriately and rigorously? 

Reviewer #1: Yes

Reviewer #2: Yes

4. Have the authors made all data underlying the findings in their manuscript fully available?

Reviewer #1: No

Reviewer #2: Yes

5. Is the manuscript presented in an intelligible fashion and written in standard English?

Reviewer #1: Yes

Reviewer #2: Yes

6. Review Comments to the Author

Reviewer #1: Thank you for the opportunity to review the revised manuscript. I believe all of my concerns have been addressed. This paper adds to the literature on determining avoidable attendance to the ED.

Reviewer #2: Revised manuscript seems to be written very good english as well as addressed all comments properly.It can be accepted for publication in Plos one journal.

7. PLOS authors have the option to publish the peer review history of their article (what does this mean?). If published, this will include your full peer review and any attached files.

Reviewer #1: **Yes: **Judah Goldstein

Reviewer #2: No

---

## [Editor Report · Acceptance letter]

18 Oct 2022

PONE-D-22-17407R1 

The Safety INdEx of Prehospital On Scene Triage (SINEPOST) study: The development and validation of a risk prediction model to support ambulance clinical transport decisions on-scene. 

Dear Dr. Miles:

I'm pleased to inform you that your manuscript has been deemed suitable for publication in PLOS ONE. Congratulations! Your manuscript is now with our production department. 

Kind regards, 

on behalf of

Dr. Yong-Hong Kuo 

Academic Editor

PLOS ONE